# Global Invasion Potential of Black-Headed and Red-Headed Webworm, *Hyphantria cunea* (Drury) (Lepidoptera: Erebidae: Arctiidae) Following Climatic Niche Simulations

**DOI:** 10.3390/insects16080843

**Published:** 2025-08-15

**Authors:** Jie Pan, Fan Shao, Jia Liu, Dongxiao Xu, Gaosheng Liu

**Affiliations:** 1College of Forestry, Nanjing Forestry University, Nanjing 210037, China; shaofan187@163.com (F.S.); jialiu1019@163.com (J.L.); xudongxiao@njfu.edu.cn (D.X.); laugousing@163.com (G.L.); 2Co-Innovation Center for Sustainable Forestry in Southern China, Nanjing Forestry University, Nanjing 210037, China

**Keywords:** niche difference, COUE, maxent, potential distribution

## Abstract

The fall webworm, *Hyphantria cunea*, a dangerous global invasive pest, has “red-headed” and “black-headed” races with distinct ecological traits. This study used the COUE framework and Maxent models to analyze their climatic niche differences and predict global distribution suitability. Findings show substantial niche differences, with the red-headed race having greater invasive potential. Both races pose larger global threats than currently recognized, surviving in North and South America, Europe, Asia, Africa, and Australia. Asia and Europe face greater threats from the black-headed race; other regions, from the red-headed race. Prevention efforts require race-specific strategies, especially in uninvaded suitable habitats.

## 1. Introduction

Invasive non-native species (invasive species, henceforth) have significant impacts on the global economy, with substantial costs incurred annually worldwide. Recent research indicates they also affect global ecosystems by altering spatial patterns and qualities of cross-ecosystem movements [1,2]. In recent years, due to increasing global race size and tighter trade connections, the number of alien invasive species has escalated, exacerbating both current and emerging biological invasions [3]. Therefore, strengthening the prevention and management of invasive species is crucial. Early Detection and Rapid Response (EDRR) strategies are considered effective in addressing biological invasion issues and minimizing the consequences of invasive species [4]. Among these strategies, risk assessment and distribution prediction of invasive species are regarded as the initial and most critical steps in invasive species prevention and management [5]. This approach can identify potential spread areas before species introduction, significantly reducing management costs and allocating resources to the most cost-effective areas, thereby providing a theoretical basis for preventing the spread of invasive species to previously unaffected regions [6,7].

The climatic niche, one of the ecological niches, serves as the foundation for risk assessment and distribution prediction of invasive species. It is also a crucial concept in determining biodiversity and biological evolution [8]. Assessing species’ climatic niches and their dynamics aids in devising effective conservation strategies for invasive species and elucidating species distributions. The dramatic changes in global climate are significantly affecting invasive species, enabling them to expand into new ranges under favorable climatic conditions [9]. Currently, many invasive species have been reported to varying extents to have expanded beyond their native climatic niches, posing greater risks to additional regions [10,11,12]. This increasingly highlights the necessity of studying the climatic niches of invasive species. Previous studies have utilized three techniques—ordination methods, ecological niche models (ENMs), and univariate methods—to quantify the dynamic changes in species’ climatic niches, with ordination and ENMs methods predominating among these approaches [13].

The ordination-based approach simplifies multivariate environmental space into lower-dimensional spaces, enabling the direct quantification of niche differences between native and invasive ranges within the same coordinate system [13]. Compared to Ecological Niche Models (ENM), its results provide a more intuitive comparison of different niches within the same environmental space and achieve more accurate quantification, albeit being more sensitive to sampling biases [13]. To address this issue, Olivier Broennimann et al. [14]. developed a COUE framework widely employed for quantifying and comparing realized niches between invasive and native ranges. This framework illustrates whether niche shifts occur during the invasion process and allows a comparison of different races within the same species [15,16]. This new framework, unlike previous methods, alleviates reliance on the fluctuating frequencies of species occurrences across diverse climatic circumstances within a region and guarantees that outcomes are unaffected by sampling attempts and environmental spatial resolution. It represents a robust method for quantifying niche differences currently available [14].

Ecological niche models (ENMs), also known as species distribution models (SDMs), are commonly used to predict the potential invasion ranges of invasive species. They calibrate predictions of potential habitat suitability using environmental data within a species’ range and its distribution [17]. Many researchers prefer various ENM models for forecasting the current and future potential habitats of pests [18,19,20]. Among these models, the MaxEnt model stands out as one of the most popular tools in the SDM field [21,22,23], widely applied to predict potential distributions of invasive organisms such as animals [24,25], and plants [26,27]. This approach enables identifying priority areas for future biological invasion management, such as anticipating Africa and Australia as priority regions for combating *Drosophila suzukii* (Matsumura) under future climate conditions [28].

The fall webworm, *Hyphantria cunea* (Drury) (Lepidoptera: Erebidae: Arctiidae), is globally recognized as a notorious invasive quarantine pest originating from North America, now extensively invading regions such as Europe and East Asia [29]. This pest primarily attacks roadside and fruit trees, feeding on over 600 plant species, particularly *Prunus domestica*, *Carya illinoinensis*, and *Malus domestica* [30,31]. In North America, there are two races: the black-headed and red-headed, exhibiting distinct food habits, biological characteristics, and ecological characteristic in both adult and larval stages. The larvae of the black-headed race have black head capsules and tend to grow on hosts in low-lying areas, such as the bottom of streams, riverbanks, and the edges of swamps. They mainly attack *Morus alba* L and *Salix*. There is no clear diurnal feeding rhythm. On the other hand, the larvae of the red-headed race have red head capsules and are more common in areas with good drainage and sparse tree canopies, such as old fields and cleared forests. They mainly attack *Carya cathayensis* Sarg., *Juglans regia* L., *Prunus maximowiczii* Rupr., and *Diospyros kaki* Thunb., and feed at night. Moreover, there are differences in the life cycles of these two types. The larval stage of the black-headed race is shorter than that of the red-headed race, while the pupal stage of the black-headed race is longer than that of the red-headed race. Their responses to photoperiodic changes are also different; the critical photoperiod for the pupal diapause of the red-headed race is longer than that of the black-headed race [32,33,34,35]. Reports indicate that the black-headed race has widely invaded Europe and East Asia, causing significant damage primarily in urban areas and feeding on the garden tree species in cities [36,37]. The red-headed race, which has not spread beyond North America, has a larger native range compared to the black-headed race, causing severe damage not only in urban areas but also across much of the continental United States [38]. Hence, we hypothesize that the climatic niche range of the red-headed *H. cunea* is broader than that of the black-headed *H. cunea*, suggesting that its potential invasion into other countries could result in more severe and extensive damage than currently observed. The current study ignores niche differences between the two races and focuses on a single invasive area compared to a native area, which can have adverse results, it is possible to underestimate or overestimate the risk areas of these two races [39,40].

In this study, the authors utilized global occurrence data of the two races of *H. cunea* as a foundation. They employed the COUE scheme and ENMs to assess niche differences between these races and their potential global distributions. The objectives of this research were: (1) To dissect climatic niche differentiation between the red-headed race and black-headed races in North America. (2) To assess the invasion potential of the Red-headed race in areas where the Black-headed race has already invaded. (3) To determine whether the invasion of the Black-headed race into Europe and Asia has resulted in a change in its climatic niche. (4) To analyze global MaxEnt modeling results for both races, delineating their potential distribution ranges 118 worldwide. These findings underscore the importance of recognizing differences between the two races of *H. cunea* and provide a scientific basis for effectively controlling the spread of both races.

## 2. Materials and Methods

### 2.1. Data Resource

#### 2.1.1. Species Occurrence Data

The occurrence records of the fall webworm within its invasive and native ranges in the United States were sourced from the Global Biodiversity Information Facility (GBIF) (https://doi.org/10.15468/dl.rnfzzz, accessed on 13 July 2025) [41], epidemic zone data released by the National Bureau of Forestry and Grassland, and articles published on Web of Science, using its recognized scientific name (*Hyphantria cunea*) as the search criterion [42,43,44,45,46]. Given that effective data in GBIF are all post–2000, the temporal range for all occurrence data was set from 2000 to 2024. Efforts were made during data collection to evenly distribute points across their respective occurrence zones, encompassing all occurrence ranges. Additionally, points with geographic coordinates uncertain beyond 5 km, erroneous coordinates, or duplicate occurrences were removed. To mitigate sampling biases, spatial thinning using the R 4.3.3 package ‘spThin’ was applied, reducing occurrence points within a 5 km radius. Ultimately, 790 occurrence points were retained for the red-headed form and 1619 for the black-headed form. Of these, 790 points for the red-headed form originated from native North American regions, while 563 points for the black-headed form were from native North American regions, 294 were from invaded European regions, and 762 were from invaded Asian regions. Detailed information on all occurrence points is provided in Figure 1.

#### 2.1.2. Climatic Data

Climatic data, including monthly average maximum temperature, average minimum temperature, and total precipitation data globally from 2000 to 2024 with a spatial resolution of 2.5 min (About 5 km × 5 km) was downloaded from WorldClim 2.1 (https://worldclim.org/data/monthlywth.html, accessed on 30 January 2024). The 19 bioclimatic variables for the period 2000–2024 averaging data across multiple years were generated using the ‘biovars’ function in the R 4.3.3 package ‘dismo’ [47] following the methodology provided by WorldClim 2.1 [48,49,50]. Ultimately, the study derived 19 bioclimatic variables representing the average conditions from 2000 to 2024, which serve as the environmental data for this research. Detailed information for each bioclimatic variable is provided in Table 1.

### 2.2. COUE Framework

COUE framework, proposed by Broennimann et al. [14]. It was employed to compare the climatic niche differences between the two races of *H. cunea* [51,52,53].

Niche overlap provides a measure of the possibility of coexistence among different races of the same species [54]. The test of ecological niche equivalence determines whether the local and introduced ecological niches are significantly equivalent by randomly redistributing the occurrence of species between the two ranges [55]. The ecological niche similarity test determines whether the local and introduced ecological niches are more similar than expected by randomly redistributing the occurrence of species within a certain range [55]. Niche overlap, equivalency tests, and similarity tests were separately generated for the two races of *H. cunea* in their native (North America) and invaded (Europe and Asia) regions. To achieve quantitative niche comparisons, occurrence points were grouped and compared for the red-headed race in North America, the black-headed race in North America, the black-headed race in Europe, and the black-headed race in Asia.

Initially, minimum convex polygons were generated around occurrence points within a 1-degree buffer zone in each range, from which background environmental data and all occurrence point environmental data were extracted. Subsequently, PCA analysis was conducted on the combined occurrence and background environmental data, focusing on the first two principal components that contributed significantly, transforming environmental variables into a grid of 100 × 100 cells two-dimensional space. Considering potential sampling biases, kernel density functions were applied to smooth occurrence densities and overall background environment, ensuring stable environmental spatial contexts for the final climatic niche quantification analysis. Schoener’s D metric was then used to quantify the magnitude of niche differentiation between the two races, ranging from 0 to 1 [56] (0 indicating no overlap, 1 indicating complete overlap). This metric was applied to compare native versus invaded ranges of the same race, different races within their native ranges, and different races in different ranges. Furthermore, Schoener’s D was used to categorize niche stability, unfilled niche space, and niche expansion, elucidating climatic niche differentiation between the two races [57]. Finally, the niche equivalence test and the niche similarity test were each conducted 1000 times utilizing the “ecospat” toolset [58]. The tests were conducted using R version 4.3.3. The tests aimed to achieve two objectives: first of all, to evaluate the equality of the ecological niches of the two races under random shuffling, and secondly, to assess whether the ecological niche overlap between the observed niche at the origin and the randomly selected niche at the invasion site surpassed the expected random value [55,59].

### 2.3. Ecological Niche Models (ENMs)

We utilized Maxent 3.4.4 software [60] to analyze data, employing 19 bioclimatic variables obtained and processed from WorldClim for the years 2000–2024 as environmental predictors. The distribution of the red-headed and black-headed races globally was partitioned into two datasets, and the model was developed at a resolution of 2.5 min. To address multicollinearity among the bioclimatic variables, a standardized procedure was applied [61,62,63].

Initially, each of the 19 bioclimatic variables was individually used in Maxent analyses for both races. Default parameters were used with 75% of the data as training sets and 25% as testing sets, repeated 10 times, and outputs were generated in “Logistic” format. Contributions of variables to predicting potentially suitable habitats were ranked based on their importance scores [64]. Subsequently, Pearson correlation analyses were performed on the 19 bioclimatic variables, selecting those with correlations ≤ 0.8 and the largest contributions (details of Pearson correlation tests and contribution rankings are provided in Figure A1, Table A1 and Table A2). After eliminating multicollinearity, the retained variables for the red-headed race are Bio 2, Bio 4, Bio 5, Bio 17, and Bio 19. The retained variables for the black-headed race are Bio 2, Bio 4, Bio 10, Bio 12, and Bio 14. Next, using the selected variables, the model was generated to simulate potentially suitable habitat ranges globally for both races under consistent settings as used during variable selection. The area under the Receiver Operating Characteristic (ROC) curve (AUC) was used as a performance metric for model evaluation [65]. AUC values range from 0 to 1, where values ≥ 0.5 indicate performance better than random predictions, and values > 0.7 indicate moderate to high model performance [66]. TSS represents the model’s ability to accurately detect the existence of both real and non-existent cases. Its value ranges from −1 to 1. The higher the value, the better the classification performance of the model [67]. The partial AUC analysis was conducted using the partial AUC program tool in R 4.3.3. This program can eliminate the AUC differences. A value greater than 1 indicates a better predictive effect [68].

## 3. Results

### 3.1. Climatic Niches Comparison Between Red-Headed and Black-Headed Races

#### 3.1.1. Univariate Comparison

The niche occupancy curves of the red-headed race compared to the black-headed race in North America (Figure 2) indicate that the niche breadth of the red-headed race exceeds that of the black-headed race across almost all environmental variables, particularly evident in bio2, bio12, bio14, bio15, bio17, and bio18. The black-headed race exhibits small expansion ranges in bio1, bio3, bio6, bio8, and bio11 variables.

#### 3.1.2. Comprehensive Comparison of Niche Differences

Comparative analysis of the niches between the red-headed and various black-headed races across different regions (Table 2 and Table 3, Figure 3). The niche overlap, unfilled index, stability index, and expansion index between the red-headed race and the North American black-headed race are 0.46, 0.111, 0.956, and 0.044, respectively. The ecological niche models for the red-headed race and the black-headed race are significantly different (*p* = 0.002), while similarity testing shows significant climate preference similarities (*p* = 0.076). This suggests substantial niche differentiation but major climate preference similarity between the North American red-headed and black-headed races, with the red-headed race exhibiting a slightly broader niche.

This indicates that there are subtle but statistically significant ecological niche differences between the red-headed and black-headed races of the fall webworm in North America. Although the magnitude of the differences is not large (mainly manifested in minor shifts in the core area and a small amount of mutual exclusion at the periphery), their systematic nature enables the overall ecological niche model to be distinguishable. This supports that the red-headed and black-headed forms are two races (or subraces) of the same species that have undergone a weak but significant ecological niche differentiation.

### 3.2. Climatic Niches Comparison Between the Native Habitat and Invasive Region of the Black-Headed Race

#### 3.2.1. Univariate Comparison

The niche occupancy curves between the native habitat of the black-headed race and its various invasive regions (Figure 4 and Figure A2) indicate that the native race generally occupies a larger niche breadth across almost all environmental variables compared to the black-headed races in both invasive regions (Europe and Asia), revealing extensive unfilled niches. Additionally, the black-headed races in the invasive regions show slight expansions in niche breadth relative to the native race. Specifically, in Asia and Europe, there are varying degrees of expansion relative to the native habitat for bio13, bio15, and bio19. The expansion is most pronounced in bio15 in Asia and bio13 in Europe. These results indicate that the black-headed race has additional suitable areas in Europe and Asia compared to its native habitat. The niche of the black-headed race has significantly changed during its invasion of Europe and Asia.

#### 3.2.2. Comprehensive Comparison of Niche Differences

Comparative niche analyses between the native and different invasive ranges of the black-headed race (Table 4 and Table 5, Figure 5, Figure A3) reveal significant insights. The niche overlap, unfilled index, stability index, and expansion index between the native range of the black-headed race and its European invasive range are 0.11, 0.508, 0.943, and 0.057, respectively. Equivalence testing indicates significant similarity in niche between these ranges (*p* = 1), while similarity testing shows dissimilar climate preferences (*p* = 0.188). This indicates that the black-headed race underwent significant actual ecological niche changes during its invasion of Europe, but there was no shift in the core ecological niche. The high unfilled index of 0.508 indicates that the native black-headed race has a broader suitable range in Europe, facilitating invasion into more extensive areas.

Additionally, the black-headed race’s niche in its native range compared to its Asian invasive range exhibits niche overlap, unfilled index, stability index, and expansion index values of 0.11, 0.211, 0.707, and 0.293, respectively. The ecological niche similarity test showed that the observed low similarity was significantly lower than the random expectation, which was in line with the expected ecological niche shift (*p* = 0.041). These indicators show that the North American black-headed race has undergone a significant ecological niche shift during its invasion of Asia, and has expanded into the new environmental space in Asia.

In conclusion, the North American black-headed race retained its original ecological niche during its invasion of Europe and did not undergo ecological niche shift. However, during its invasion of Asia, ecological niche shift occurred. Moreover, there are very large uninfested suitable areas in both Europe and Asia.

### 3.3. Potential Invasion Comparison Between the Red-Headed and Black-Headed Races Globally

The Maxent models has a high degree of reliability in predicting the potential suitable distribution ranges of the red-headed and black-headed races. This can be seen from the AUC values of the red-headed race (0.940, pAUC = 15.747, TSS = 0.876); and the AUC values of the black-headed race (0.916, pAUC = 16.107, TSS = 0.888). Model results, depicted in Figure 6 and Figure 7, show that the suitable range for the red-headed race not only encompasses much of the currently suitable areas for the black-headed race but also extends significantly beyond, covering larger suitable areas in southern South America, southern Africa, and Australia. The black-headed race maintains a considerable potential distribution range in the currently widely invaded areas of Asia and Europe, as well as in non-colonized regions. Specifically (Figure 8, Table A3), in Asia and Europe, the suitable range for the black-headed race exceeds that of the red-headed race, totaling approximately 4.7726 million square kilometers. Conversely, in North America, Australia, South America, Africa, and Oceania, the suitable range for the red-headed race exceeds that of the black-headed race, totaling approximately 9.0601 million square kilometers. This indicates that both races of *H. cunea* have broader potential suitable ranges globally compared to current ranges, posing greater potential invasion risks. Furthermore, in specific regions, the impact of the red-headed race is generally greater than that of the black-headed race.

## 4. Discussion

In this study, the niche differences between two races of *Hyphantria cunea* in their native and invasive ranges were analyzed using the COUE framework [14] and Maxent model [69]. Additionally, the global potential distribution ranges of these two races were predicted. Comparisons were made to illustrate the niche differences between the red-headed race and different regional races of the black-headed race, the niche differences between the native range of the black-headed race and various invasive regions, and the potential distribution ranges of both races globally. This analysis aims to elucidate the niche differentiation between the red-headed and black-headed races, the niche shift in the black-headed race during invasion, and the invasive risks and potential distribution ranges of both red-headed and black-headed races globally.

The ecological niches of the red-headed and black-headed races of the fall webworm were compared. It was found that there were subtle but statistically significant differences in their ecological niches, and the ecological niche of the red-headed race was slightly wider than that of the black-headed race. This indicates that, compared to the black-headed race, the red-headed race may cause damage over a wider range. This finding is consistent with previous observations; in North America, the red-headed race is widely distributed from west to east in most areas, spanning all the states of the United States, while the black-headed race is limited to the central and eastern regions of the country and is not observed to be distributed in the west [34]. As early as 1973, Ito, Y. and Warren, L. proved the differences in feeding preference, behavior, and nesting structure between these races [70]. These results emphasize the importance of conducting separate studies on the red-headed and black-headed races. Therefore, in future analyses of the potential distribution or risk assessment of the fall webworm, it is necessary to independently analyze these two races. Moreover, the ecological niche width of the red-headed race exceeds that of the black-headed race, indicating that if the red-headed race can maintain its ecological niche width to a certain extent, if the red-headed race invades regions such as Europe and Asia, it may occupy a larger environmental space and cause significant damage.

By comparing the climatic ecological niches of the black-headed race in its native habitat with those in different invaded areas, we observed the ecological niche shifts during the invasion process in Asia through single-variable and comprehensive ecological niche difference analyses. During the invasion into Europe, the black-headed race underwent a significant actual ecological niche change (contraction), but no core ecological niche shift occurred. This is consistent with previous research results, namely that the black-headed fall webworm (*H. cunea*) in the European invasion retained its native ecological niche, with an extremely small or almost zero expansion index, and exhibited a large number of unfilled areas of the native ecological niche [39]. Additionally, genomic studies have shown that the fall webworm, due to its strong environmental adaptability and evolutionary capabilities, gradually evolved into different races during its invasion in Asia [71,72]. These findings collectively suggest a high likelihood of climatic niche shifts occurring during the invasion of other countries by the black-headed *H. cunea*.

Based on Maxent modeling, we further conducted a global-scale prediction of potentially suitable ranges for two races of *H. cunea*. According to Figure 6, Figure 7 and Figure 8, both races of *H. cunea* exhibit broader potential suitable ranges globally relative to current distributions, with the red-headed race generally posing greater potential risks than the black-headed race in certain regions. In comparison with Ge et al.’s use of Climex to predict the global suitable ranges of *H. cunea*, our predicted suitable ranges were less extensive [73]. This difference may stem from the differing principles underlying the two models. Several studies indicate that Climex consistently produces broader predictions than Maxent [74,75,76]. Climex operates by simulating species survival under appropriate climatic conditions given a set of parameters, potentially reflecting the subjective biases of the modelers and leading to predictions that exceed those of the Maxent model. Maxent, in contrast, predicts species probability distributions based on the principle of maximum entropy, providing more objective results but susceptible to biases from distributional data points. To minimize subjective influences, we ultimately opted to use Maxent for separate prediction analyses of these two races [77].

One interesting extension of this study is that the red-headed race’s slightly wider ecological niche corresponds to a much larger potential suitable range in North America, South America, and Australia compared to the black-headed race. Logically, this should cause greater and more extensive harm than the black-headed race. However, the red-headed race is only distributed in North America, and there are no relevant records in other regions of the world. Only the black-headed race has widely invaded regions such as Europe and Asia. This might be due to the wide spread of the black-headed fall webworm (*H. cunea*), which might have been affected, prompting an increase in global management efforts for this species, regardless of its races. Additionally, the netting of the black-headed race is thinner and it spreads harm to trees after the fifth instar, while the netting of the red-headed race is more obvious. The larvae of the red-headed race stay in the netting during the day and leave it at night to forage [33,78]. The larval period of the black-headed race is shorter than that of the red-headed race, and the pupal period of the black-headed race is longer than that of the red-headed race [34]. This makes the black-headed race more likely to be a main carrier of human transmission and less likely to be detected, making it easier to spread to other regions with goods. The red-headed race is more likely to be detected, which is not conducive to its spread. However, regions such as South America, Africa, and Australia that have not experienced the spread of the fall webworm yet might face a greater invasion risk due to the lack of prior prevention experience for this pest. Therefore, it is necessary to strengthen the monitoring of the fall webworm in the already-invaded Asian and European regions and other potential suitable areas to prevent the red-headed race from causing harm in these potential suitable areas. Future efforts should focus on strengthening the monitoring of both races of fall webworm to mitigate their further global expansion and related risks.

Our study was also limited by only considering climatic niche differences and employed the Maxent model exclusively for predicting suitable habitat ranges. Factors such as plant species closely associated with insect growth were not incorporated into our analysis, particularly concerning folivorous pests. Additionally, integrated ecological niche modeling is widely recognized for yielding more accurate predictive outcomes [79,80]. Future research could encompass multiple models and factors related to the pest, including climate and plant variables, utilizing diverse ecological niche modeling approaches. This approach could provide decision-makers with more detailed and effective strategies for preventing and managing invasive outbreaks of *H. cunea*.

## 5. Conclusions

In this study, we found that the niche range of the red-headed race of *H. cunea* is significantly broader than that of the black-headed race, indicating its potential for causing more extensive harm compared to the black-headed race. The niche of the black-headed race of *H. cunea* changed during its invasion into Asia, while minimal changes were observed during its invasion into Europe. Despite the current occurrence of the red-headed species being confined to North America, its climatic potential distribution shows greater invasive potential in multiple regions compared to the black-headed species, particularly in South America, Australia, and Africa. Both races of pests have extensive potential suitable ranges globally, with large portions of suitable habitats remaining unfilled.

## Figures and Tables

**Figure 1 insects-16-00843-f001:**
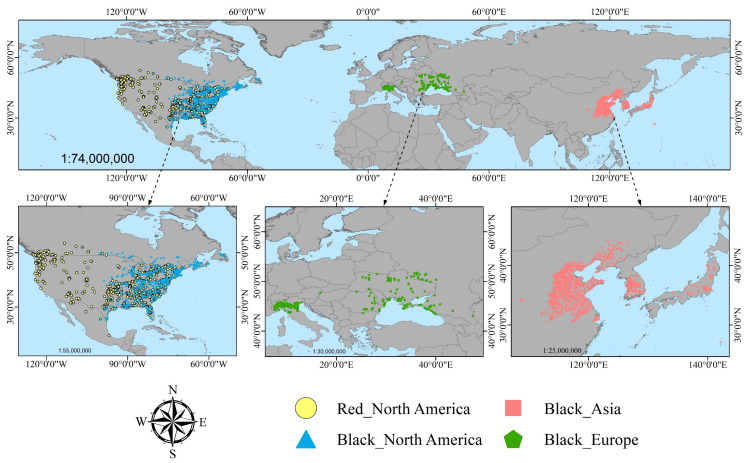
Global occurrence map of the two races of the fall webworm, *Hyphantria cunea*. (Yellow points represent the red-headed race in North America, blue points represent the black-headed race in North America, pink points represent the black-headed race in Asia, and green points represent the black-headed race in Europe).

**Figure 2 insects-16-00843-f002:**
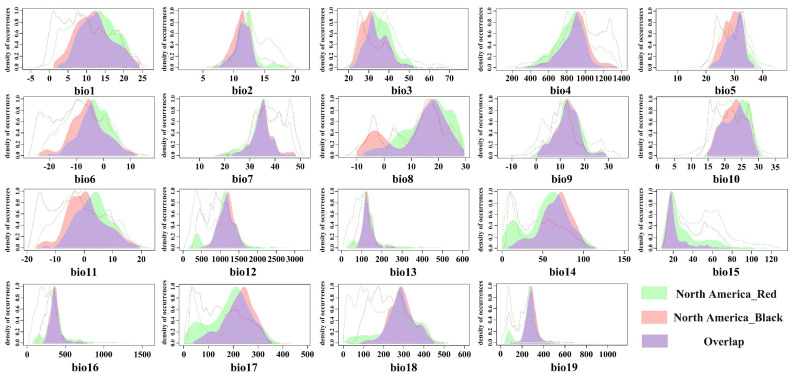
The kernel density drawing of bioclimatic variables between the red-headed and black-headed races in North America. (Colors representing the Native niche indicate bioclimatic variables specific to the red-headed race; colors indicating the Invasive niche represent bioclimatic variables unique to the black-headed race in North America; colors indicating the Niche overlap represent bioclimatic variables shared between the red-headed and black-headed races in North America).

**Figure 3 insects-16-00843-f003:**
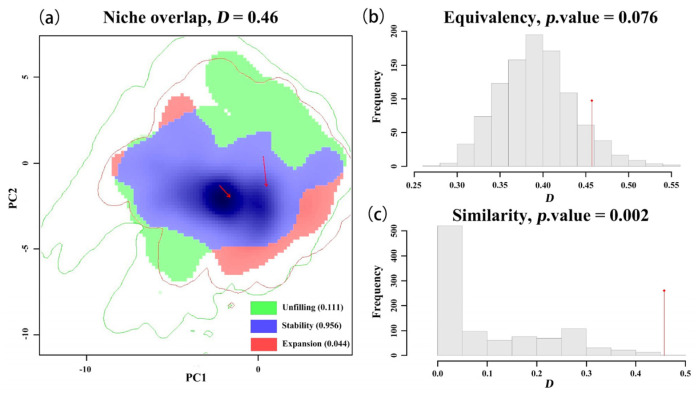
Comprehensive drawing of niche overlap, equivalence, and similarity tests between the red-headed and black-headed races in North America. (**a**) Niche overlap. The solid arrow points to the center of mass, indicating the direction from the native ecological niche of the species to the invasive ecological niche. (**b**) equivalence test. (**c**) similarity test. The histogram represents the zero distribution of D obtained from 1000 iterations, and it is compared with the observed Schoener’s D index (red diamond).

**Figure 4 insects-16-00843-f004:**
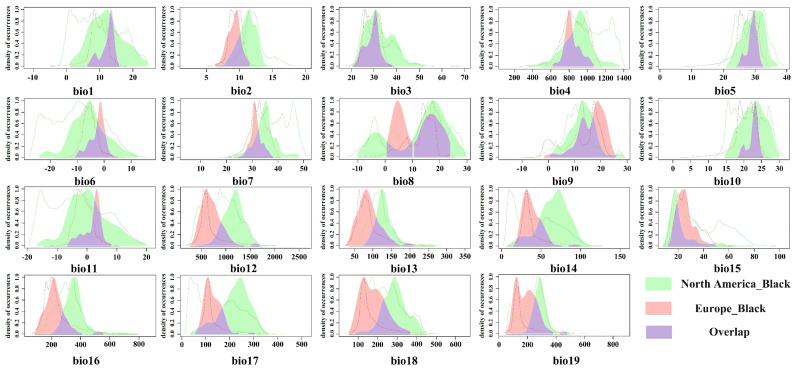
The kernel density drawing of bioclimatic variables between the origin and the Europe of black-headed races. (The colors corresponding to the Native niche in the figure represent the bioclimatic variable ranges specific to the black-headed race in North America; the colors corresponding to the Invasive niche represent the bioclimatic variable ranges specific to the black-headed race in Europe; colors indicating Niche overlap represent the bioclimatic variable ranges shared by the black-headed race in both North America and Europe).

**Figure 5 insects-16-00843-f005:**
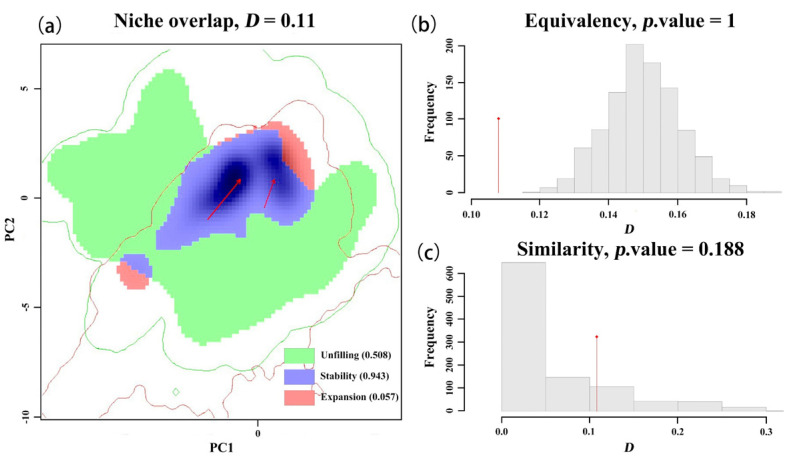
Comprehensive drawing of niche overlap, equivalence, and similarity tests between the native habitat of the black-headed race and its invasive range in Europe. (**a**) Niche overlap. The solid arrow points to the center of mass, indicating the direction from the native ecological niche of the species to the invasive ecological niche. (**b**) equivalence test. (**c**) similarity test. The histogram represents the zero distribution of D obtained from 1000 iterations, and it is compared with the observed Schoener’s D index (red diamond).

**Figure 6 insects-16-00843-f006:**
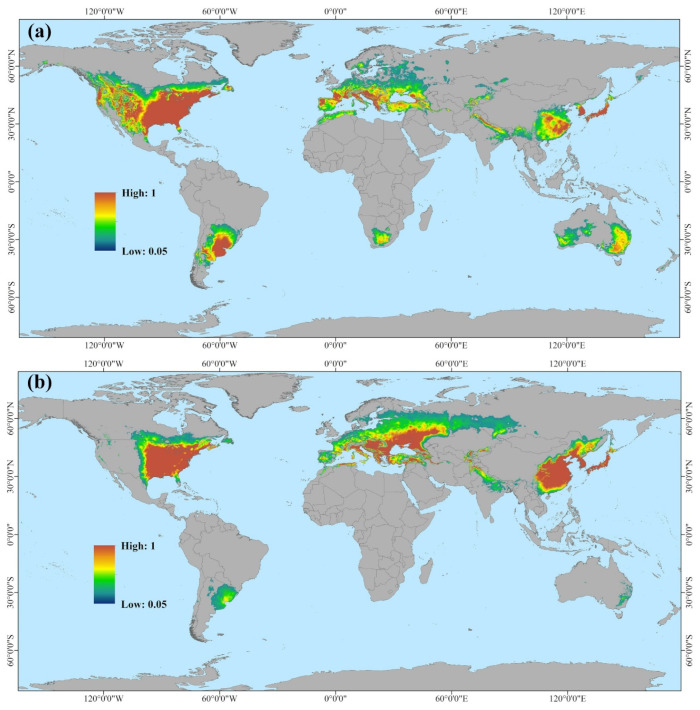
The potential suitable ranges and suitability levels of the two races of *Hyphantria cunea* globally. ((**a**) Red-headed; (**b**) Black-headed).

**Figure 7 insects-16-00843-f007:**
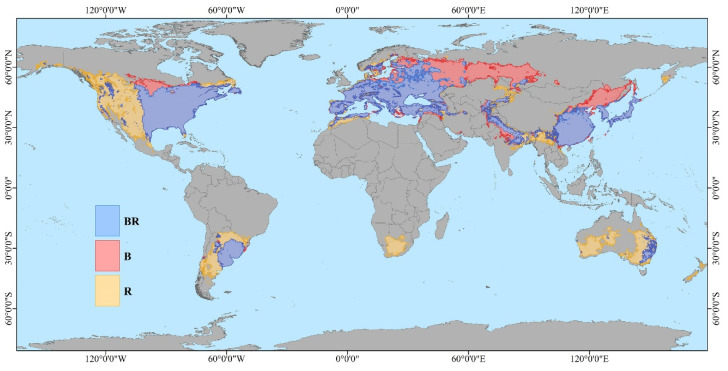
The global distribution dynamics of the two races of *Hyphantria cunea*. (The color range corresponding to letter B in the figure represents the exclusive global potential suitable habitat for the black-headed race. The color range corresponding to the letter R in the figure represents the exclusive global potential suitable habitat for the red-headed race. The color range corresponding to the letter BR in the figure represents the globally shared potential suitable habitat for both red-headed and black-headed races).

**Figure 8 insects-16-00843-f008:**
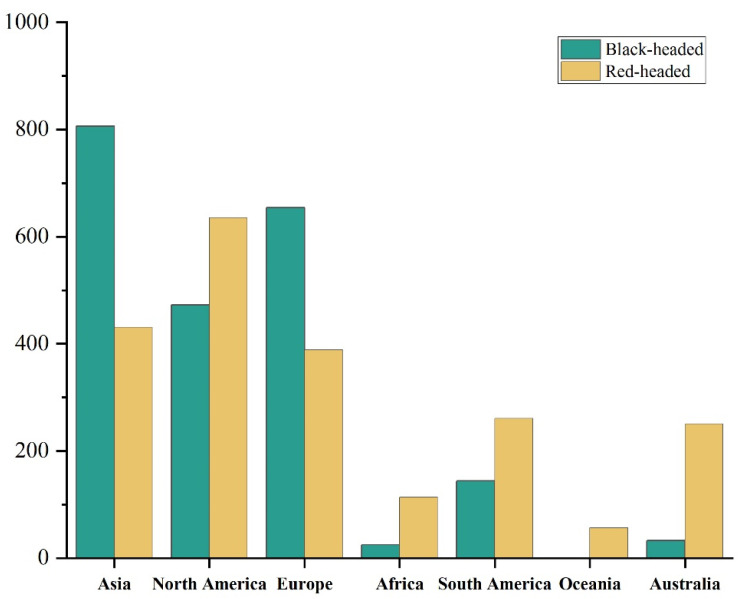
The difference in suitable habitat areas for the two races of *Hyphantria cunea* across different continents worldwide. (Unit: ×10^4^ square kilometers).

**Table 1 insects-16-00843-t001:** Environmental variables specific content.

Variables	Description
bio1 (°C)	Annual mean temperature
bio2 (°C)	Mean diurnal range (mean of monthly (max temp—min temp)
bio3	Isothermality (bio2/bio7) (×100)
bio4	Temperature seasonality (standard deviation × 100)
bio5 (°C)	Max temperature of the warmest month
bio6 (°C)	Min temperature of coldest month
bio7 (°C)	Temperature annual range (bio5–bio6)
bio8 (°C)	Mean temperature of wettest quarter
bio9 (°C)	Mean temperature of driest quarter
bio10 (°C)	Mean temperature of warmest quarter
bio11 (°C)	Mean temperature of coldest quarter
bio12 (mm)	Annual precipitation
bio13 (mm)	Precipitation of wettest month
bio14 (mm)	Precipitation of driest month
bio15 (mm)	Precipitation seasonality (coefficient of variation)
bio16 (mm)	Precipitation of wettest quarter
bio17 (mm)	Precipitation of driest quarter
bio18 (mm)	Precipitation of warmest quarter
bio19 (mm)	Precipitation of coldest quarter

**Table 2 insects-16-00843-t002:** Niche overlap testing, niche similarity testing, and niche equivalence testing between the red-headed race and black-headed races in North America. (RH stands for red-headed race; BH stands for black-headed race).

	RH→BH in North America
Schoener’s D	0.46
Similarity test *p* value	0.002
Equivalency test *p* value	0.076

**Table 3 insects-16-00843-t003:** Expansion index, stability index, and unfilled index comparisons between the red-headed race and black-headed races in North America. (RH stands for red-headed race; BH stands for black-headed race).

	RH→BH in North America
Unfilling	0.111
Stability	0.956
Expansion	0.044

**Table 4 insects-16-00843-t004:** Niche overlap testing, niche similarity testing, and niche equivalence testing between the native habitat of the black-headed race and its various invasive ranges.

	**North America→Europe**	**North America→Asia**
Schoener’s D	0.11	0.11
Similarity test *p* value	0.188	0.329
Equivalency test *p* value	1	0.041

**Table 5 insects-16-00843-t005:** Expansion index, stability index, and unfilled index comparisons between the native habitat of the black-headed race and its various invasive ranges.

	**North America→Europe**	**North America→Asia**
Unfilling	0.508	0.211
Stability	0.943	0.707
Expansion	0.057	0.293

## Data Availability

The original contributions presented in the study are included in the article, further inquiries can be directed to the corresponding author.

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
