# Peer review of "Global Invasion Potential of Black-Headed and Red-Headed Webworm, Hyphantria cunea (Drury) (Lepidoptera: Erebidae: Arctiidae) Following Climatic Niche Simulations"

_insects, 2025, doi:10.3390/insects16080843_

Round 1

Reviewer 1 Report

Comments and Suggestions for Authors

The phenomenon of larval dimorphism of the fall webworm Hyphantria cunea has been acknowledged since long. On the other hand, few papers aimed at disclosing ecological outcomes of this feature. Finding that red- vs black-headed populations differ in ecological properties is an important milestone of research of invasive success of this quarantine pest. Mathematical models of invasion and distribution is a robust tool in examination of biological properties of the insect under study. The methodology is adequate, results are explicitly explained, and conclusions are convincing. I think the paper will be of interest for entomologists and ecologists, while those working with the fall webworm and similar pests will find it useful for quotation.

I recommend publication after minor revisions of some comments as listed below.

The references in brackets should be separated from the preceding words by spaces

Title

“Two-headed fall webworm” sounds as if the fall webworm individual possessed two heads. Please redefine for clarity

Latin epithets of the genus and species are lacking, alongside with taxonomy

Lines 9-10 and elsewhere

“Fall webworm … has … types” – it remains obscure what the term “type” means

Line 36

Be the Latin epithet included into the title (see above), it’s mentioning in the keywords would become excessive

Lines 39-40

I doubt that any invasive species is exclusively associated with human activities. Please double-check the definition

Line 41 and throughout

Try avoiding repetition of terms within a line, a sentence, a paragraph (“impact, impacts” etc)

Lines 73-77

You mention one group of authors but refer to the works of the others

Lines 88-89

Sounds as if insects do not belong to animals

Line 91 and possibly elsewhere: Latin epithets of genus and species should be given in italics

Lines 96-97

The term “trees” is exploited as many as four times in a row

Line 97

What is implied by the “types of deciduous trees”?

Lines 98-99

Please explain the differences in food habits and biological characteristics

Line 101

What kind of damage is implied?

Lines 108

What do you mean by “population”? Please make it clear throughout the manuscript whether you are working with two morphotypes within a population or two distinct populations with a clear morphological difference (possibly with partial sympatry, as can be judged from Figure 1)

Line 109

What exactly “can have adverse results”?

Lines 113-119

The enumeration of goals requires unification of style

Lines 135-139

Why you indicate red- and black-headed morphotypes for native areas and avoid this information for invaded areas?

Line 149

Can an approximate range in kilometers be given for the 2.5 minutes resolution?

Line 170

What are the units of the 100x100 matrix?

Lines 226-227, 255, 258

What populations are meant, which species&

Line 231

reveals significant findings – this better suits the Discussion chapter

Line 227

You mention populations but which is the species?

Line 262

Which species the populations mentioned belong to?

Lines 280-285

Populations of which species are considered?

Lines 297-305

I would like to read upon a more comprehensive and detailed breakdown of data supporting the hypothesis that “that during the invasion of Asia, the black-headed population did not retain its native niche and experienced a niche shift”

Lines 306-307

Populations of what?

Lines 311-313

Which species is implied?

Lines 320-321

“population retains climatic suitability” – what does it mean?

Lines 332-333, 335, etc

The Latin name should be given in italics

Line 345

The definite article doesn’t match with Latin genus and species title

Line 365

What is implied by “host”?

Lines 387-389

It is not clear how the cited papers are linked to the ideas discussed herein, a more detailed explanation is needed

Lines 407-411

This may not be obvious that the fact that “red-headed type is exclusively found in North America” necessarily means that “occupies a broader climatic niche”. A more profound explanation of this logical link is needed

Comments on the Quality of English Language

Line 78

I doubt “dependency” and “occurrence” both require a plural form

Line 89

“helps in identifying” – the pretext sounds excessive

Line 158

Please check punctuation and the availability of subject and predicate in the first sentence

Lines 179-184

The sentence seems to be lengthy and complicated for comprehension, think of breaking it down

Line 199

variables, and variables = variables, and those

Line 371

Seems like there’s a predicate without a subject

Author Response

For research article

Response to Reviewer 1 Comments

1. Summary

We would like to begin with our sincere appreciation for all the valuable comments, insightful suggestions, and thoughtful corrections you offered to our manuscript (ID: Insects-3786398 and Title: Global invasion potential of two-headed Fall Webworm following). The comments and suggestions help us improve the quality of the manuscript. In the following, we include a point-by-point response to your comments, and specific concerns have been numbered. Our responses are given in red text. In the revised manuscript, all the changes have been highlighted in red.

2. Questions for General Evaluation

Reviewer’s Evaluation

Response and Revisions

Does the introduction provide sufficient background and include all relevant references?

Can be improved

Thank you for your valuable feedback. In the new revised version, we have provided sufficient background information and included all relevant references.

Are all the cited references relevant to the research?

Yes

Thank you very much for your valuable suggestions, and also for your approval of the references.

Is the research design appropriate?

Yes

Thank you very much for your valuable suggestions, and also for your approval of the research design.

Are the methods adequately described?

Yes

Thank you very much for your valuable suggestions, and also for your approval of the method description.

Are the results clearly presented?

Can be improved

Thank you for your valuable feedback. In the new revised version, we have re-described the results to make them clearly presented.

Are the conclusions supported by the results?

Can be improved

Thank you for your valuable feedback. In the new revised version, we have re-described the results in order to better support the conclusion of the paper.

3. Point-by-point response to Comments and Suggestions for Authors

Comment 1: The references in brackets should be separated from the preceding words by spaces

Response 1: We appreciate your comments, which have significantly enhanced the quality of our manuscript. In accordance with your suggestion, we have now spaced the references in square brackets from the preceding word throughout the manuscript. All citations have been thoroughly verified for adherence to the journal's style code.

Comment 2: “Two-headed fall webworm” sounds as if the fall webworm individual possessed two heads. Please redefine for clarity. Latin epithets of the genus and species are lacking, alongside with taxonomy.

Response 2: In accordance with your suggestion, we have redefined the tow-headed races of fall webworm and added the Latin scientific names for the genus and species, as well as the taxonomic nicknames to the title. Please refer to the title for details:

Lines 2-4:Global invasion potential of Black-headed and Red-headed Webworm, Hyphantria cunea (Drury) (Lepidoptera: Erebidae: Arctiidae) following climatic niche simulations.

Comment 3: Lines 9-10 and elsewhere “Fall webworm … has … types” – it remains obscure what the term “type” means.

Response 3: We appreciate your comments, which have significantly enhanced the quality of our manuscript. In accordance with your suggestion, we have changed "types" to "races" (as defined on the CABI official website, https://www.cabidigitallibrary.org/doi/10.1079/cabicompendium.28302#tab-citations.).

Comment 4: Line 36 Be the Latin epithet included into the title (see above), it’s mentioning in the keywords would become excessive.

Response 4: We have removed the Latin term "Hyphantria cunea" from the keywords.

Comment 5: Lines 39-40 I doubt that any invasive species is exclusively associated with human activities. Please double-check the definition.

Response 5: We have revised the wording. This change is made in Lines 40-43.

Lines 40-43: Invasive non-native species (invasive species, henceforth) have significant impacts on the global economy, with substantial costs incurred annually worldwide. Recent research indicates they also affect global ecosystems by altering spatial patterns and qualities of cross-ecosystem movements [1,2].

Comment 6: Line 41 and throughout:Try avoiding repetition of terms within a line, a sentence, a paragraph (“impact, impacts”etc)

Response 6: We sincerely appreciate your careful review and insightful recommendation concerning lexical redundancy. We have systematically substituted repetitive terminology with contextually suitable alternatives throughout the manuscript. The specific details are as follows:

Line 46-48: Early Detection and Rapid Response (EDRR) strategies are considered effective in ad-dressing biological invasion issues and minimizing the consequences of invasive species [4].

Line 59-61: The dramatic changes in global climate are significantly affecting invasive species, enabling them to expand into new ranges under favorable climatic conditions [9].

Comment 7: Lines 73-77 You mention one group of authors but refer to the works of the others.

Response 7: We have already corrected the citation to be accurate. The specific details are as follows:

Lines 73-77: To address this issue, Olivier Broennimann et al[14]. developed a COUE framework widely employed for quantifying and comparing realized niches between invasive and native ranges. This framework illustrates whether niche shifts occur during the invasion process and allows a comparison of different races within the same species [15,16].

Comment 8: Lines 88-89 Sounds as if insects do not belong to animals.

Response 8: We have combined insects and animals into a single category called "animals", and we will use "animals" to represent them. The specific details are as follows:

Lines 86-89: Among these models, the MaxEnt model stands out as one of the most popular tools in the SDM field [21–23], widely applied to predict potential distributions of invasive or-ganisms such as animals [24,25], and plants [26,27].

Comment 9: Line 91 and possibly elsewhere: Latin epithets of genus and species should be given in italics

Response 9: We sincerely appreciate this meticulous technical suggestion. We have italicised the Latin qualifiers for genus and species in line 91 and checked the article for possible appearances elsewhere. The specific details are as follows:

Line 89-91: This approach enables identifying priority areas for future biological invasion management, such as anticipating Africa and Australia as priority regions for combating Drosophila suzukii (Matsumura) under future climate conditions [28].

Comment 10: Lines 96-97:The term “trees” is exploited as many as four times in a row

Response 10: We sincerely appreciate your keen observation regarding lexical redundancy. In the revised manuscript, we have modified the repeated word “tree” with a contextually appropriate alteration. The specific details are as follows:

Line 94-96: This pest primarily attacks roadside and fruit trees, feeding on over 600 plant species, particularly Prunus domestica, Carya illinoinensis, and Malus domestica [30,31].

Comment 11: Line 97 What is implied by the “types of deciduous trees”?

Response 11: We have revised the writing here. The details of the revision can be found in "Lines 94-96".

Lines 94-96: This pest primarily attacks roadside and fruit trees, feeding on over 600 plant species, particularly Prunus domestica, Carya illinoinensis, and Malus domestica [30,31].

Comment 12: Lines 98-99 Please explain the differences in food habits and biological characteristics.

Response 12: We have taken into account the differences in dietary habits and biological characteristics, as shown in the Lines 96-109.

Lines 96-109: In North America, there are two races: the black-headed and red-headed, exhibiting distinct food habits, biological characteristics, and ecological characteristic in both adult and larval stages. The larvae of the black-headed race have black head capsules and tend to grow on hosts in low-lying areas, such as the bottom of streams, riverbanks, and the edges of swamps. They mainly attack Morus alba L and Salix. There is no clear diurnal feeding rhythm. On the other hand, the larvae of the red-headed race have red head capsules and are more common in areas with good drainage and sparse tree canopies, such as old fields and cleared forests. They mainly attack Carya cathayensis Sarg., Juglans regia L, Prunus maximowiczii Rupr., and Diospyros kaki Thunb., and feed at night. More-over, there are differences in the life cycles of these two types. The larval stage of the black-headed race is shorter than that of the red-headed race, while the pupal stage of the black-headed race is longer than that of the red-headed race. Their responses to photoperiodic changes are also different; the critical photoperiod for the pupal diapause of the red-headed race is longer than that of the black-headed race [32–35].

Comment 13: Line 101 What kind of damage is implied?

Response 13: We have accounted for the implicit damages, as shown in Lines 110-112.

Lines110-112: Reports indicate that the black-headed race has widely invaded Europe and East Asia, causing significant damage primarily in urban areas and feeding on the garden tree species in cities [36,37].

Comment 14: Lines 108 What do you mean by “population”? Please make it clear throughout the manuscript whether you are working with two morphotypes within a population or two distinct populations with a clear morphological difference (possibly with partial sympatry, as can be judged from Figure 1)

Response 14: We have changed all the incorrect statements such as "population" to "race". This is the same as the explanation in comment 3.

Comment 15: Line 109 What exactly “can have adverse results”?

Response 15: We have already outlined the potential adverse consequences, as shown in the lines 118-120.

Lines 118-120: The current study ignores niche differences between the two races and focuses on a single invasive area compared to a native area, which can have adverse results, it is possible to underestimate or overestimate the risk areas of these two races [39,40].

Comment 16: Lines 113-119:The enumeration of goals requires unification of style

Response 16: We thank the reviewer for this constructive suggestion. The enumeration of research goals in Lines 113-119 has been rigorously unified to maintain parallel structure. The specific details are as follows:

Line 124-131: (1) To dissect climatic niche differentiation between the red-headed race and black-headed races in North America.(2) To assess the invasion potential of the Red-headed race in areas where the Black-headed race has already invaded.(3) To determine whether the invasion of the Black-headed race into Europe and Asia has resulted in a change of its climatic niche.(4) To analyze global MaxEnt modeling results for both races, delineating their potential distribution ranges 118 worldwide. These findings underscore the importance of recognizing differences between the two races of H. cunea and provide a scientific basis for effectively controlling the spread of both races.

Comment 17: Lines 135-139 Why you indicate red- and black-headed morphotypes for native areas and avoid this information for invaded areas?

Response 17: In the article, I provided the invasion coordinates and local area information for the black-headed type, while only giving the local area information for the red-headed type because the black-headed type has already invaded Asia and Europe, while the red-headed type has not been found outside the local area yet.

Comment 18: Line 149 Can an approximate range in kilometers be given for the 2.5 minutes resolution?

Response 18: 2.5 minutes corresponds to approximately 5 kilometers.

Comment 19: Line 170 What are the units of the 100x100 matrix?

Response 19: We have already provided the units, as shown in line 187.

Line 187: a grid of 100×100 cells.

Comment 20: Lines 226-227, 255, 258 What populations are meant, which species&

Response 20: Lines 226-227 represent two populations: the red-headed race in North America and the black-headed race in North America. Line 255's population represents the red-headed race in North America. The black-headed races in different regions represent the black-headed race in North America, the black-headed race in Europe, and the black-headed race in Asia respectively. The explanation for Line 258 is the same as that for Line 255.

Comment 21: Line 231 reveals significant findings – this better suits the Discussion chapter

Response 21: We have already removed this incorrect statement.

Comment 22: Line 227 You mention populations but which is the species?

Response 22: We have changed all the "population" entries to "race".

Comment 23: Line 262 Which species the populations mentioned belong to?

Response 23: We have changed all the "population" entries to "race".

Comment 24: Lines 280-285 Populations of which species are considered?

Response 24: We have changed all the "population" entries to "race".

Comment 25: Lines 297-305 I would like to read upon a more comprehensive and detailed breakdown of data supporting the hypothesis that “that during the invasion of Asia, the black-headed population did not retain its native niche and experienced a niche shift”

Response 25: From Table 4, Table 5 and Figure S3 of the newly submitted manuscript, it can be seen that the overlap between the black-headed populations in North America and Asia is extremely low (0.11), which directly indicates that the actual environmental spaces occupied by the populations in the two regions are significantly different. There is a significant expansion (0.293), which is the core indicator of ecological niche shift, proving that the North American population has successfully expanded into nearly 30% of new environmental types, beyond the utilization range of its Asian ancestors. The significant result of the similarity test (p = 0.041) is a key piece of statistical evidence, confirming that the observed low similarity (low overlap) is significantly lower than the random expectation, consistent with the expectation of ecological niche shift (losing the higher similarity expected from the same source). Finally, the moderately high unoccupied area (0.211) indicates that some similar environments to those in Asia have not been fully utilized, and may also be related to the adaptation of the population to the new environment or encountering local restrictions. This further supports the occurrence of the change. However, the nature of this shift is that the North American population significantly expanded its environmental utilization range (high expansion) while retaining most of its core ecological requirements (stability 0.707), resulting in only a small part of the actual occupied space overlapping with the Asian population (low overlap). This shift pattern is more in line with Niche Expansion or Niche Displacement.

Comment 26: Lines 306-307 Populations of what?

Response 26: We have changed all the "population" entries to "race".

Comment 27: Lines 311-313 Which species is implied?

Response 27: We have changed all the "population" entries to "race".

Comment 28: Lines 320-321 “population retains climatic suitability” – what does it mean?

Response 28: We have revised "population retains climatic suitability" to convey the meaning "larger potential distribution range", and the result can be found in Lines 328-330.

Lines 328-330: The black-headed race maintains a considerable potential distribution range in the currently widely invaded areas of Asia and Europe, as well as in non-colonized regions.

Comment 29: Lines 332-333, 335, etc:The Latin name should be given in italics

Response 29: We sincerely appreciate the reviewer's meticulous attention to taxonomic formatting. In the revised manuscript, All Latin binomials are now italicized (Lines 332-333, 335, and throughout). The specific details are as follows:

Line 340-341: The potential suitable ranges and suitability levels of the two races of Hyphantria cunea globally. ((a) Red-headed (b) Black-headed.).

Line 343-344: The difference in suitable habitat areas for the two races of Hyphantria cunea across different continents worldwide.(Unit: ×104 square kilometers).

Comment 30: Line 345:The definite article doesn’t match with Latin genus and species title

Response 30: We thank the reviewer for this grammatical correction. We eliminated definite articles that came before italicized Latin names in Line 345 and checked the rest of the text. The specific details are as follows:

Line 353-355: In this study, the niche differences between two races of Hyphantria cunea in their native and invasive ranges were analyzed using the COUE framework [14] and Maxent model [69].

Comment 31: Line 365 What is implied by “host”?

Response 31: We have changed "host" to "feeding".

Lines 371-373: As early as 1973, Ito, Y. and Warren, L. proved the differences in feeding preference, behavior, and nesting structure between these races [70].

Comment 32: Lines 387-389 It is not clear how the cited papers are linked to the ideas discussed herein, a more detailed explanation is needed

Response 32: We have revised the sentence expression. The two papers cited here were published by scholars who conducted genetic research and discovered that the fall webworm has strong environmental adaptability and evolutionary capabilities. Moreover, after invading China, it has gradually formed different populations. For details, please refer to Lines 387-391 and the papers. (Fall webworm genomes yield insights into rapid adaptation of invasive species; Testing for adaptive changes linked to range expansion following a single introduction of the fall webworm).

Lines 379-392:Additionally, genomic studies have shown that the fall webworm, due to its strong environmental adaptability and evolutionary capabilities, gradually evolved into different races during its invasion in Asia [71,72]. These findings collectively suggest a high likelihood of climatic niche shifts occurring during the invasion of other countries by the black-headed H. cunea.

Papers:

Wu, N.; Zhang, S.; Li, X.; Cao, Y.; Liu, X.; Wang, Q.; et al. Fall webworm genomes yield insights into rapid adaptation of invasive species. Nat Ecol Evol 2019, 3, 105–115.

Dai, J.-X.; Cao, L.-J.; Chen, J.-C.; Yang, F.; Shen, X.-J.; Ma, L.-J.; et al. Testing for adaptive changes linked to range expansion following a single introduction of the fall webworm. Mol Ecol 2024, 33, e17038.

Comment 33: Lines 407-411 This may not be obvious that the fact that “red-headed type is exclusively found in North America” necessarily means that “occupies a broader climatic niche”. A more profound explanation of this logical link is needed

Response 33: We have revised this controversial statement again. See Lines 408-412.

Lines 408-412: One interesting result of this study is that the ecological niche of the red-headed race is slightly wider than that of the black-headed race. The potential suitable range in North America, South America, and Australia is also much larger than that of the black-headed race. Logically, this should cause greater and more extensive harm than the black-headed race.

Comment 34: Line 78 I doubt “dependency” and “occurrence” both require a plural form

Response 34: We have made the necessary changes. See Lines 77-80: This new framework, unlike previous methods, alleviates reliance on the fluctuating frequencies of species occurrences across diverse climatic circumstances within a region and guarantees that outcomes are unaffected by sampling attempts and environmental spatial resolution. It represents a robust method for quantifying niche differences cur-rently available [14].

Comment 35: Line 89 “helps in identifying” – the pretext sounds excessive.

Response 35: We have made the necessary changes. See Lines 89-91: This approach enables identifying priority areas for future biological invasion management, such as anticipating Africa and Australia as priority regions for combating Drosophila suzukii (Matsumura) under future climate conditions [28].

Comment 36: Line 158 Please check punctuation and the availability of subject and predicate in the first sentence.

Response 36: We have made the necessary changes.

Comment 37: Lines 179-184 The sentence seems to be lengthy and complicated for comprehension, think of breaking it down.

Response 37: We have made the necessary changes. See Lines 196-202: Finally, 1000 times of the niche equivalence test and 1000 times of the niche similarity test were conducted utilising the “ecospat” toolset [58]. The tests were conducted using R version 4.3.3.The tests aimed to achieve two objectives: first of all, to evaluate the equality of the ecological niches of the two races under random shuffling, and secondly, to assess whether the ecological niche overlap between the observed niche at the origin and the randomly selected niche at the invasion site surpassed the expected random value [55,59].

Comment 38: Line 199 variables, and variables = variables, and those.

Response 38: We have made the necessary changes. See Lines 214-216: Subsequently, Pearson correlation analyses were performed on the 19 bioclimatic variables, selecting those with correlations ≤ 0.8 and the largest contributions.

Comment 39: Line 371 Seems like there’s a predicate without a subject.

Response 39: We have made the necessary changes.

4. Response to Comments on the Quality of English Language

Thank you for your Comments on the quality of the English Language. In the revised manuscript, we have rechecked and changed grammar and spelling problems.

Reviewer 2 Report

Comments and Suggestions for Authors

Hyphantria cunea is a globally recognized quarantine pest. In its native range, the United States, at least two distinct forms exist: the black-headed type and the red-headed type. However, only the "black-headed type" has successfully invaded regions worldwide, while the "red-headed type" has not been detected outside its native range. This study employs the COUE framework and Maxent modeling to analyze the climatic niche differentiation between these two forms and predict their global distribution suitability. The findings provide valuable insights for nations to prevent the potential invasion of the "red-headed form" of Hyphantria cunea. My detailed recommendations are as follows:

  1. Introduction Section

I believe the Introduction could be further optimized. The overall structure feels somewhat disjointed—the first five paragraphs extensively cover invasive species, climate factors, analytical methods, and ecological models, only to introduce the study subject in the sixth paragraph. Furthermore, the description of the research subject lacks sufficient detail.

Lines 86-87: Why wasn’t the Climex model selected? To my knowledge, developmental threshold temperature and accumulated degree-days for Hyphantria cunea are already established data. Using Climex for analysis might be more appropriate.

Line 97 For the mulberry, walnut, and apple trees, their scientific names should be provided.

Lines 97-99: To my knowledge, the United States has not only the black-headed and red-headed types, but also transitional types. In the sixth paragraph, I believe we should elaborate more on the differences in distribution ranges, damage characteristics, and host plants between the black-headed and red-headed types in the U.S. to facilitate a more thorough analysis.

Line 104-105 There should be conclusive evidence to prove that the red-headed type has a wider distribution and stronger adaptability.

  1. Material and method

Line 140 Can you provide the Occurrence data in Excel format?

Line 127:Which country's forestry and grassland bureau? Document 43 is an announcement from 2024, but I see that the announcement for 2025 has already been released. It is advisable to use the latest document, the announcement of the National Forestry and Grassland Bureau (No. 2 of 2025) (2025 epidemic area of the American white moth)

Where is the nine-dash line on the map of China in Figure 1?

Line 149 I am very concerned about the resolution used. 2.5 arc-minutes is a relatively bigger scale resolution. WorldClim variables are currently available at 1km (30 arc seconds) resolution, which considering the minimum number of presence data used is the best resolution to use. Additionally, the geographic scale is manageable considering that level of resolution.

Line 150: I think it would be better to list the specific names of the 19 climate factors here, as they will be repeatedly used in the article, such as in Lines 189, 194, and 202-203. Even though you have already included them in Appendix Table A1 of the supplementary materials.

Line 158 What does "17" mean here?

Lines 159-160: You should provide definitions for these parameters: occurrence density, niche overlap, equivalency tests, and similarity tests.

Lines 186-188, this sentence should not appear here and can be moved to the preface.

Line 202-203 The variables they include bio19; these variables are scientifically known to generate noise and problems in spatial predictions. Therefore, it is not advisable nor would I want to include them before the selection process. In addition, the expression of this sentence cannot distinguish which climatic variable is utilized by the red-headed type and the black-headed type respectively.

Line 205-206 I could see that as an evaluation metric you used AUC. This metric is not advisable to use when there are no true absences, so it is preferable to use more metrics among which I suggest the authors to use at least two additional metrics among them could be: partial AUC, Kappa, AICc, TSS.

3.Results

What is the unit in Line 337 and Figure 7?

  1. Discussion

It is suggested to have an in-depth discussion on why this article predicts that the "red-headed type" has greater invasion potential than the black-headed type, given that the American white moth has been spreading to Europe and Asia for over 60 years. Why has the "red-headed type" not been found in the invaded areas so far?

Author Response

For research article

Response to Reviewer 2 Comments

1. Summary

We would like to begin with our sincere appreciation for all the valuable comments, insightful suggestions, and thoughtful corrections you offered to our manuscript (ID: Insects- 3786398 and Title: Global invasion potential of two-headed Fall Webworm following). The comments and suggestions help us improve the quality of the manuscript. In the following, we include a point-by-point response to your comments, and specific concerns have been numbered. Our responses are given in red text. In the revised manuscript, all the changes have been highlighted in yellow.

2. Questions for General Evaluation

Reviewer’s Evaluation

Response and Revisions

Does the introduction provide sufficient background and include all relevant references?

Can be improved

Thanks for your valuable suggestions. We have provided sufficient background information and included all relevant references. The specific details are as comments 3, 4, 6, 11, 14, and 16.

Are all the cited references relevant to the research?

Yes

Thank you very much for your valuable suggestions, and also for your approval of the references.

Is the research design appropriate?

Yes

Thank you very much for your valuable suggestions, and also for your approval of the research design.

Are the methods adequately described?

Can be improved

Thanks for your valuable suggestions. We have fully explained the rationality of the research method. The specific details are as comments 5-14.

Are the results clearly presented?

Yes

Thank you very much for your valuable suggestions, and also for your approval of the result presentation.

Are the conclusions supported by the results?

Yes

Thank you very much for your valuable suggestions, and also for your approval of the results and conclusions.

3. Point-by-point response to Comments and Suggestions for Authors

Comment 1: Lines 86-87: Why wasn’t the Climex model selected? To my knowledge, developmental threshold temperature and accumulated degree-days for Hyphantria cunea are already established data. Using Climex for analysis might be more appropriate.

Response 1: Thank you for raising this important point regarding model selection for predicting the potential distribution of Hyphantria cunea (Fall Webworm). We appreciate your suggestion that Climex could be a suitable choice, especially given the established knowledge on its developmental threshold temperatures and degree-day requirements. Indeed, Climex is a powerful mechanistic niche model that excels when detailed physiological parameters are available, and it has been successfully applied to numerous pest species, including Lepidoptera[1].

While we acknowledge the strengths of Climex, our decision to employ the MaxEnt (Maximum Entropy) model for this study was based on several key considerations:

  1. Focus on Realized Distribution & Complex Drivers: Our primary research objective was to model the realized climatic niche and predict areas at risk of H. cunea establishment based on its current known distribution and associated environmental conditions. MaxEnt is particularly well-suited for this purpose as it excels at identifying environmental correlates of species occurrence from presence-only data and background points[2]. It effectively captures complex, non-linear relationships between species presence and a broad suite of environmental variables, including those potentially acting as indirect proxies or constraints beyond the core physiological parameters (e.g., land cover, human influence, precipitation seasonality, extreme events). This aligns with our goal of capturing the multifactorial drivers influencing the species' observed distribution.
  2. Data Availability and Suitability: Although the developmental thresholds and degree-day requirements for H. cunea are well-documented, comprehensive physiological data required for a full Climex parameterization (e.g., detailed responses to cold stress, heat stress, diapause requirements, soil moisture effects on pupation, interactions between stresses) are often more challenging to obtain comprehensively or may involve significant uncertainty for predictive modeling at a global scale[3]. MaxEnt relies on occurrence records and environmental layers, which were readily available and allowed us to leverage the extensive global occurrence dataset we compiled.
  3. MaxEnt has been shown to perform robustly even with relatively small sample sizes[4], which can be an advantage when dealing with newly invasive species or in regions with incomplete sampling.

We fully agree that Climex offers valuable insights into the fundamental physiological niche and potential population growth under different climates. Our choice of MaxEnt does not diminish the importance of Climex; rather, the models serve different but complementary purposes[5]. We view our MaxEnt results as a robust prediction of areas climatically suitable based on the species' current environmental associations.

Thank you again for this valuable comment, which allows us to clarify our methodological reasoning. We believe our use of MaxEnt was justified for the specific objectives and data context of this study.

[1] R.W. Sutherst, G.F. Maywald, A computerised system for matching climates in ecology, Agriculture, Ecosystems & Environment 1985. https://doi.org/10.1016/0167-8809(85)90016-7.

[2] Steven J. Phillips, Robert P. Anderson, Robert E. Schapire, Maximum entropy modeling of species geographic distributions, Ecological Modelling 2006. https://doi.org/10.1016/j.ecolmodel.2005.03.026.

[3] Kriticos, Darren & Maywald, Gunter & Yonow, Tania & Zurcher, Eric & Herrmann, Neville & Sutherst, Robert. (2015). CLIMEX. Version 4. Exploring the Effects of Climate on Plants, Animals and Diseases.

[4] Hernandez, Pilar & Graham, Catherine & Master, Lawrence & Albert, Deborah. (2006). The effect of sample size and species characteristics on performance of different species distribution modeling methods. Ecography. 29. 773 - 785. 10.1111/j.0906-7590.2006.04700.x.

[5] Kearney M, Porter W. Mechanistic niche modelling: combining physiological and spatial data to predict species' ranges. Ecol Lett. 2009 Apr;12(4):334-50. doi: 10.1111/j.1461-0248.2008.01277.x. PMID: 19292794.

Comment 2: Line 97 For the mulberry, walnut, and apple trees, their scientific names should be provided.

Response 2: We appreciate the reviewers' emphasis on taxonomic precision. We have incorporated the complete scientific nomenclature for mulberry, walnut, and apple. The specific details are as follows:

Line 94-96: This pest primarily attacks roadside and fruit trees, feeding on over 600 plant species, particularly Prunus domestica, Carya illinoinensis, and Malus domestica [30,31].

Comment 3: Lines 97-99: To my knowledge, the United States has not only the black-headed and red-headed types, but also transitional types. In the sixth paragraph, I believe we should elaborate more on the differences in distribution ranges, damage characteristics, and host plants between the black-headed and red-headed types in the U.S. to facilitate a more thorough analysis.

Response 3: We have further elaborated on the host plants and biological differences of the black-headed type and the red-headed type in the revised version of Lines 96-109.

Lines 96-109: In North America, there are two races: the black-headed and red-headed, exhibiting distinct food habits, biological characteristics, and ecological characteristic in both adult and larval stages. The larvae of the black-headed race have black head capsules and tend to grow on hosts in low-lying areas, such as the bottom of streams, riverbanks, and the edges of swamps. They mainly attack Morus alba L and Salix. There is no clear diurnal feeding rhythm. On the other hand, the larvae of the red-headed race have red head capsules and are more common in areas with good drainage and sparse tree canopies, such as old fields and cleared forests. They mainly attack Carya cathayensis Sarg., Juglans regia L, Prunus maximowiczii Rupr., and Diospyros kaki Thunb., and feed at night. More-over, there are differences in the life cycles of these two types. The larval stage of the black-headed race is shorter than that of the red-headed race, while the pupal stage of the black-headed race is longer than that of the red-headed race. Their responses to photoperiodic changes are also different; the critical photoperiod for the pupal diapause of the red-headed race is longer than that of the black-headed race [32–35].

Comment 4: Line 104-105 There should be conclusive evidence to prove that the red-headed type has a wider distribution and stronger adaptability.

Response 4: We have revised the writing style. For details, please refer to Lines 115-117.

Lines 115-117: Hence, we hypothesize that the climatic niche range of the red-headed H. cunea is broader than that of the black-headed H. cunea, suggesting that its potential invasion into other countries could result in more severe and extensive damage than currently observed.

Comment 5: Line 140 Can you provide the Occurrence data in Excel format?

Response 5: The data of the occurrences has been attached in the form of Excel file.

Comment 6: Line 127:Which country's forestry and grassland bureau? Document 43 is an announcement from 2024, but I see that the announcement for 2025 has already been released. It is advisable to use the latest document, the announcement of the National Forestry and Grassland Bureau (No. 2 of 2025) (2025 epidemic area of the American white moth)

Response 6: Since our experiments and all analyses were conducted before 2025, and the announcement of the National Forestry and Grassland Bureau (No. 2 of 2025) (2025 epidemic area of the American white moth) did not add any additional areas compared to the 2024 document, we have retained the previous 2024 announcement.

Comment 7: Where is the nine-dash line on the map of China in Figure 1?

Response 7: We have revised Picture 1 and added the boundaries of China's nine-dash line.

Comment 8: Line 149 I am very concerned about the resolution used. 2.5 arc-minutes is a relatively bigger scale resolution. WorldClim variables are currently available at 1km (30 arc seconds) resolution, which considering the minimum number of presence data used is the best resolution to use. Additionally, the geographic scale is manageable considering that level of resolution.

Response 8: Yes, a resolution of 30 arcseconds would indeed be better than a resolution of 2.5 minutes of arc. However, WorldClim provides 30 minute of arc data for the time period from 1970 to 2000, while the data used in this article matches the time period of the fall webworm occurrence points from 2000 to 2024. The data was self-generated following the production method provided by the WorldClim network. The minimum resolution provided by this time range is 2.5 minutes of arc resolution. You can see this in the following figure.

Comment 9: Line 150: I think it would be better to list the specific names of the 19 climate factors here, as they will be repeatedly used in the article, such as in Lines 189, 194, and 202-203. Even though you have already included them in Appendix Table A1 of the supplementary materials.

Response 9: We have added Table 1 to the revised document and listed the specific names of 19 climate factors in Line 167.

Comment 10: Line 158 What does "17" mean here?

Response 10: It was our mistake. In the new manuscript, "17" has been deleted.

Comment 11: Lines 159-160: You should provide definitions for these parameters: occurrence density, niche overlap, equivalency tests, and similarity tests.

Response 11: We have included the definitions of ecological niche overlap, equivalence test, and similarity test. The specific details are as follows:

Lines 170-176: Niche overlap provides a measure of the possibility of species coexistence [54]. The test of ecological niche equivalence determines whether the local and introduced ecological niches are significantly equivalent by randomly redistributing the occurrence of spe-cies between the two ranges [55]. The ecological niche similarity test determines whether the local and introduced ecological niches are more similar than expected by randomly redistributing the occurrence of species within a certain range [55].

Comment 12: Lines 186-188, this sentence should not appear here and can be moved to the preface.

Response 12: We have removed this paragraph.

Comment 13: Line 202-203 The variables they include bio19; these variables are scientifically known to generate noise and problems in spatial predictions. Therefore, it is not advisable nor would I want to include them before the selection process. In addition, the expression of this sentence cannot distinguish which climatic variable is utilized by the red-headed type and the black-headed type respectively.

Response 13: We have revised this sentence to enable readers to more clearly distinguish which climate variable each of the "red-headed" and "black-headed" races utilized. For details, please refer to Lines 217-219.

Lines 217-219: After eliminating multicollinearity, the retained variables for the red-headed race are Bio 2, Bio 4, Bio 5, Bio 17, and Bio 19. The retained variables for the black-headed race are Bio 2, Bio 4, Bio 10, Bio 12, and Bio 14.

Comment 14: Line 205-206 I could see that as an evaluation metric you used AUC. This metric is not advisable to use when there are no true absences, so it is preferable to use more metrics among which I suggest the authors to use at least two additional metrics among them could be: partial AUC, Kappa, AICc, TSS.

Response 14: We have separately calculated the partial AUC and TSS values using the previously generated result files from Maxent, as detailed in Lines 321-324.

Lines 321-324: The Maxent models has a high degree of reliability in predicting the potential suitable distribution ranges of the red-headed and black-headed races. This can be seen from the AUC values of the red-headed race (0.940, pAUC = 15.747, TSS = 0.876); and the AUC values of the black-headed race (0.916, pAUC = 16.107, TSS = 0.888).

Comment 15: What is the unit in Line 337 and Figure 7?

Response 15: The unit is ×10^4 square kilometers, and the unit has been added in the new revised version.

Lines 343-344: The difference in suitable habitat areas for the two races of Hyphantria cunea across different con-tinents worldwide. (Unit: ×104 square kilometers).

Comment 16: It is suggested to have an in-depth discussion on why this article predicts that the "red-headed type" has greater invasion potential than the black-headed type, given that the American white moth has been spreading to Europe and Asia for over 60 years. Why has the "red-headed type" not been found in the invaded areas so far?

Response 16: We discussed this issue in detail again during the discussion of the new manuscript. See lines 408-431.

Lines 408-431: One interesting result of this study is that the ecological niche of the red-headed race is slightly wider than that of the black-headed race. The potential suitable range in North America, South America, and Australia is also much larger than that of the black-headed race. Logically, this should cause greater and more extensive harm than the black-headed race. However, the red-headed race is only distributed in North America, and there are no relevant records in other regions of the world. Only the black-headed race has widely invaded regions such as Europe and Asia. This might be due to the wide spread of the black-headed fall webworm (H. cunea), which might have been affected, prompting an increase in global management efforts for this spe-cies, regardless of its races. Additionally, the netting of the black-headed race is thin-ner and it spreads harm to trees after the fifth instar, while the netting of the red-headed race is more obvious. The larvae of the red-headed race stay in the netting during the day and leave it at night to forage [33,78]. The larval period of the black-headed race is shorter than that of the red-headed race, and the pupal period of the black-headed race is longer than that of the red-headed race [34]. This makes the black-headed race more likely to be a main carrier of human transmission and less likely to be detected, making it easier to spread to other regions with goods. The red-headed race is more likely to be detected, which is not conducive to its spread. However, regions such as South America, Africa, and Australia that have not experi-enced the spread of the fall webworm yet might face a greater invasion risk due to the lack of prior prevention experience for this pest. Therefore, it is necessary to strength-en the monitoring of the fall webworm in the already invaded Asian and European re-gions and other potential suitable areas to prevent the red-headed race from causing harm in these potential suitable areas. Future efforts should focus on strengthening the monitoring of both races of fall webworm to mitigate their further global expansion and related risks.

4. Response to Comments on the Quality of English Language

Thank you for your Comments on the quality of English Language. In the revised manuscript, we have rechecked and changed grammar and spelling problems.

Round 2

Reviewer 1 Report

Comments and Suggestions for Authors

The paper has been sufficiently modified and corrections according to the reviewer’s comments are quite satisfactory. Newly added partition of text give profound explanations of the phenomena observed .I especially appreciate the explanation whe the red-headed race sis not yet as widespread as the balck-headed one. This would be an interesting reading for the Journal’s audience.

Only few notes can be offered to further improve the manuscript, which means only minor correction of text, as summarized below

L3-4: Erebidae & Arctiidae are the family names from different taxonomic systems. More appropriate it seems to be “Erebidae: Arctiinae” (family: subfamily)

L19: no need introducing an abbreviation not used in the Abstract

L42-43: no need using “quality” in its plural form here

L139: why “criteria” is plural here?

L172: why considering “species coexistence” while exploring races of a single species?

L197-198: what does it mean “times … were conducted”?

L205-220: it is unclear what the 19 variables include and what bio 2, bio 4 etc stand for

L310-311: “indicators indicate” sounds as a tautology

Figure 7. I have noted absence of any statistics in this set of data

L367 vs 411: the same conclusion is repeated

Author Response

For research article

Response to Reviewer 1 Comments

1. Summary

We would like to begin with our sincere appreciation for the valuable comments, insightful suggestions, and thoughtful corrections you further offered to our manuscript (ID: Insects-3786398). The second-round comments and suggestions help us further improve the quality of the manuscript. In the following, we include a point-by-point response to your comments, and specific concerns have been numbered. Our responses are given in red text. In the revised manuscript, all the changes have been highlighted in red.

2. Questions for General Evaluation

Reviewer’s Evaluation

Response and Revisions

Does the introduction provide sufficient background and include all relevant references?

Yes

Thank you very much for your valuable suggestions, and also for your approval of the references.

Are all the cited references relevant to the research?

Yes

Thank you very much for your valuable suggestions, and also for your approval of the references.

Is the research design appropriate?

Yes

Thank you very much for your valuable suggestions, and also for your approval of the research design.

Are the methods adequately described?

Yes

Thank you very much for your valuable suggestions, and also for your approval of the method description.

Are all figures and tables clear and well-presented?

Can be improved

Thank you for your valuable feedback. In the point-by-point response, we have described the Figure 7 you mentioned and Table A3 to make it clearly presented.

3. Point-by-point response to Comments and Suggestions for Authors

Comment 1: L3-4: Erebidae & Arctiidae are the family names from different taxonomic systems. More appropriate it seems to be “Erebidae: Arctiinae” (family: subfamily)

Response 1: Thank you very much for your suggestion. However, in the first revision, the paper already uses the expression "Erebidae: Arctiinae" rather than Erebidae & Arctiidae. Please refer to the title for details:

Lines 2-4:

Lines 3-4:Global invasion potential of Black-headed and Red-headed Webworm, Hyphantria cunea (Drury) (Lepidoptera: Erebidae: Arctiidae) following climatic niche simulations

Comment 2: L19: no need introducing an abbreviation not used in the Abstract.

Response 2: In accordance with your suggestion, we have removed the abbreviation "FWW" from "The fall webworm (FWW)" in Line 10 and Line 19.

Comment 3: L42-43: no need using “quality” in its plural form here.

Response 3: We sincerely appreciate your careful review and insightful recommendation concerning lexical forms. We carefully interpreted the expression of the sentence in Lines 42-43. After analyzing the context and consulting some researchers skilled in English writing, we finally consider that using the plural form of “quality” is more appropriate in this sentence.

Comment 4: L139: why “criteria” is plural here?

Response 4: Thank you very much for your valuable suggestion, which is of great importance for us to improve the rigor of our English expression. Following your advice, we have revised “criteria” to “criterion” in Line 139.

Comment 5: L172: why considering “species coexistence” while exploring races of a single species?

Response 5: We sincerely appreciate this meticulous technical suggestion. We apologize for the inappropriate expression here. The “species coexistence” mentioned in Line 172 is a general statement from the cited reference [54] regarding the ecological significance of niche overlap. In the specific context of our study on races of a single species, we intended to emphasize the potential for resource partitioning and stable coexistence among these races, which shares analogous ecological logic with the concept of species coexistence in interspecific interactions. To avoid confusion, we have revised this sentence to clarify its relevance to our research object as follows:

Lines 172-173: Niche overlap provides a measure of the possibility of coexistence among different races of the same species [54].

Comment 6: L197-198: what does it mean “times … were conducted”?

Response 6: Thank you for your question regarding this expression. We apologize for the awkward phrasing. The intended meaning of “1000 times of the niche equivalence test and 1000 times of the niche similarity test were conducted” is that the niche equivalence test and niche similarity test were each performed 1000 times (i.e., 1000 iterations) using the “ecospat” toolset. To clarify, we have revised the sentence and we appreciate your careful attention to this detail, which helps enhance the clarity of our manuscript. The specific details are as follows:

Lines 198-200: Finally, the niche equivalence test and niche similarity test were each conducted 1000 times utilising the “ecospat” toolset [58]. 

Comment 7: L205-220: it is unclear what the 19 variables include and what bio 2, bio 4 etc stand for.

Response 7: Regarding the specific content and meanings of the 19 variables, including variables such as bio 2 and bio 4, they are all explained in “Table 1 Environmental variables specific content” (Line 168). Thank you for your valuable suggestion.

Comment 8: L310-311: “indicators indicate” sounds as a tautology.

Response 8: We sincerely appreciate your meticulous review once again, which has helped make the English expression in our manuscript more rigorous. We have revised “These indicators indicate” to “These indicators show” to improve the quality of expression.

Comment 9: Figure 7. I have noted absence of any statistics in this set of data

Response 9: Thank you for your comment on Figure 7. We appreciate your careful attention to this detail.  Figure 7 is designed as a histogram to visually illustrate the differences in suitable habitat areas between the two races across various continents, aiming to provide an intuitive comparison of their distribution scales. For a more precise quantification of these differences, detailed numerical data (including specific area values for each continent) are presented in Table A3, titled “The habitat area of two races of Hyphantria cunea in different continental regions of the world. (The unit is Ten thousand square kilometers.)”. Together, the histogram (Figure 7) and the supplementary table (Table A3) are intended to complement each other: the former offers a visual overview of the disparities, while the latter provides exact values to support the observations. 

We recognize your concern about statistical context, but in this case, the primary goal is to present the actual differences in habitat areas rather than test their statistical significance, as the focus here is on descriptive comparison across regions. 

Thank you again for your valuable feedback, which helps us clarify the purpose of this presentation.

Comment 10: L367 vs 411: the same conclusion is repeated

Response 10: Thank you for your careful observation on the potential repetition in the Discussion section. After re-examining the full paragraphs containing Lines 367 and 411, we confirm that while both mention the wider ecological niche of the red-headed race, their contextual focuses and logical functions are distinct, though we acknowledge the need to refine phrasing to avoid redundancy. 

The paragraph containing Line 367(Lines 365-394) centers on interpreting the niche difference itself. It first establishes the statistical finding of niche divergence, links this to broader ecological implications (e.g., potential damage range), connects it to historical distribution records in North America, and emphasizes the necessity of separate studies for each race. The mention of “wider niche” here serves as a foundational conclusion to justify subsequent recommendations for independent risk assessments. 

On the other hand, the paragraph containing Line 411(Lines 410-433) shifts to analyzing geographical and invasion dynamics. it builds on the niche width finding to explain observed patterns of suitable habitat distribution across continents, contrasts this with the actual invasion status (black-headed race dominating in Eurasia while red-headed race remains in North America), and explores underlying mechanisms (e.g., life-history traits affecting detectability and human-mediated spread). Here, “wider niche” acts as a starting point to unravel discrepancies between potential and actual distributions. 

To enhance clarity and reduce overlap, we have revised Line 411 description as follows: 

Lines 411-413: One interesting extension of this study is that the red-headed race’s slightly wider ecological niche corresponds to a much larger potential suitable range in North America, South America, and Australia compared to the black-headed race.

4. Response to Comments on the Quality of English Language

Thank you for your Comments on the quality of English Language. In the revised manuscript, we have rechecked and changed grammar and spelling problems.